# Histological Lesions and Replication Sites of PCV3 in Naturally Infected Pigs

**DOI:** 10.3390/ani11061520

**Published:** 2021-05-24

**Authors:** Elisa Rigo De Conti, Talita Pilar Resende, Lacey Marshall-Lund, Albert Rovira, Fabio Augusto Vannucci

**Affiliations:** 1Veterinary School, Federal University of Rio Grande do Sul, Porto Alegre 91540-000, Brazil; elisa.rdc@hotmail.com; 2Veterinary Diagnostic Laboratory, College of Veterinary Medicine, University of Minnesota, 1333 Gortner Avenue, Saint Paul, MN 55108, USA; leml@umn.edu (L.M.-L.); rove0010@umn.edu (A.R.); 3Department of Veterinary and Biomedical Sciences, College of Veterinary Medicine, University of Minnesota, 1971 Commonwealth Avenue, Saint Paul, MN 55108, USA; talitaresendevet@gmail.com

**Keywords:** PCV3, in situ hybridization, histopathology

## Abstract

**Simple Summary:**

Diagnosing porcine circovirus type 3 (PCV3) is a challenge in pig production. Although the virus has been recently isolated, the patterns of PCV3-associated histological lesions are still to be elucidated. The present study describes the association of PCV3 mRNA by in situ hybridization within histological lesions and PCV3 DNA detected by real-time PCR in naturally infected pigs. The main histologic lesions associated with PCV3 mRNA detection were lymphoplasmacytic myocarditis and lymphoplasmacytic interstitial pneumonia, in heart and lung, respectively. Our findings offer robust guidance of microscopic lesions associated with PCV3, which may have a key role in PCV3 diagnosis.

**Abstract:**

Porcine circovirus type 3 (PCV3) has been recently described as a potential cause of abortions and systemic vasculitis in pigs. Although the virus has been detected by real-time PCR in several porcine tissues from countries worldwide, PCV3-associated diseases have not been satisfactorily clarified. The objective of this study was to investigate the association between the presence of PCV3 mRNA detected by in situ hybridization (ISH) within histological lesions and PCV3 DNA detected by real-time PCR in naturally infected pigs. A total of 25 PCV3 PCR-positive cases were analyzed. Formalin-fixed tissues from these cases were evaluated for histologic lesions and for ISH-RNA positive signals for PCV3. The most frequent tissue type with histopathologic lesions was heart, 76.2%, with lymphoplasmacytic myocarditis and epicarditis as the most frequent lesions observed. Lymphoplasmacytic interstitial pneumonia was also a frequent finding, 47.6%. There were also lesions in kidney, liver, spleen and lymph nodes. PCV3-ISH-RNA positive signals were mostly observed in association with lymphoplasmacytic inflammatory infiltrate in various tissues, including arteries. Based on our results, the minimum set of specimens to be submitted for histopathology and mRNA in situ hybridization to confirm or exclude a diagnosis of PCV3 are heart, lung and lymphoid tissues (i.e., spleen and lymph nodes), especially for differential diagnosis related with PCV2-associated diseases.

## 1. Introduction

A novel porcine circovirus, porcine circovirus type 3 (PCV3), was identified in 2016 in the US [1,2]. After initial identification, PCV3 has also been reported in several countries in Asia, Europe and South America [3,4,5,6,7,8,9,10]. PCV3-infected pigs have shown clinical signs and lesions compatible with porcine dermatitis and nephropathy syndrome, reproductive failure, and cardiac and multi-systemic inflammation [1,2,11,12], clinical signs that can be also present in pigs infected with porcine circovirus type 2 (PCV2) [13].

Porcine circoviruses have been a concern in the pig industry since early 1990s [14]. PCV2 is a well-known worldwide endemic virus in pig farms and is the etiologic agent of porcine circovirus associated diseases (PCVAD) [15]. There are other circoviruses that can infect pigs. PCV1 has not shown clinical importance in swine production [16] and had less than 80% nucleotide sequence identity with PCV2 [17]. Recently, PCV4 has been described in China, showing 43.2% of nucleotide sequence identity with PCV3, but its clinical importance has yet to be elucidated [18]. Although PCV2 and PCV3 belong to the Circoviridae family, their genetic similarity at whole genome level is only 48%, and they share only 25–37% of similarity in the open reading frame 2 (ORF2) [1]. The genetically closest viral genome to PCV3 in the public database belongs to a bat-feces’ associated circovirus, with replicase and capsid proteins showing 55 and 35% identity to PCV3, respectively [1].

The lack of an experimental model to elucidate the role of PCV3 in pig diseases is still a major issue. Attempts to reproduce PCV3 disease under experimental conditions without the virus isolation has been made. Using an infectious PCV3 DNA clone, typical clinical signs resembling those of PDNS-like disease were observed in specific-pathogen-free (SPF) piglets experimentally challenged with PCV3 [12]. In another study, using a tissue homogenate PCV3-positive as inoculum, no clinical sings were found in the cesarean-derived, colostrum-deprived (CD/CD) inoculated pigs [11]. However, the histological evaluation of those pigs demonstrated lesions consistent with multisystemic inflammation and perivasculitis, which suggest a role of PCV3 in subclinical infection [11]. Similar data was found in CD/CD pigs inoculated with PCV3 obtained from virus isolation [19]. Even with no clinical sings observed, pigs presented histological lesions consistent with multi systemic inflammation characterized by myocarditis and systemic perivasculitis, which corroborates a subclinical PCV3 infection [19].

The detection of PCV3 mRNA in association with histological lesions by in situ hybridization (ISH-RNA) represents the presence of actively replicating virus within the affected tissues, suggesting the primary involvement of the virus in the pathogenesis of the lesions observed [1]. ISH-RNA has been an important tool to identify the presence of pathogens within histological lesions, especially for emerging pathogens, in which isolation of infectious agents has not been accomplished or is laborious, and when there are no commercial antibodies available for immunohistochemistry [1,20,21,22,23]. Hence, the development of comprehensive diagnostic criteria including detection of the virus in association with histological lesions is critical for guiding diagnostic investigation, and also for understanding the pathogenesis of the PCV3-associated diseases. The aim of this study was to investigate the distribution of PCV3 mRNA detected by in situ hybridization ISH-RNA within histological lesions and its association with Ct values of real-time PCR.

## 2. Materials and Methods

### 2.1. Case Selection

Cases for this study were selected from cases routinely submitted to the University of Minnesota Veterinary Diagnostic Laboratory (UMN-VDL) for diagnostic investigation from 2016 to 2018. Wasting, dyspnea, weight loss and diarrhea were the main concerns related by the field veterinarians. Cases were selected based on tissue homogenate positive for PCV3 by real-time PCR where the diagnostic report suggested a potential involvement of PCV3 in the clinical outcome. The cases were selected for histopathology reevaluation and duplex mRNA in situ hybridization (ISH-RNA) for PCV3 and PCV2. As the diversity of tissues in each case were dependent on the submitter, only cases that had at least three formalin-fixed sections of the following organs were selected: heart, spleen, lung, kidney, liver and/or lymph nodes.

### 2.2. Histopathology and ISH-RNA

Hematoxylin and eosin (H&E)-stained tissues were evaluated for histopathological lesions by two independent blinded pathologists. The lesions in each tissue were recorded, compiled and organized according to the organ (Table 1).

ISH-RNA was performed as described previously [1,24] using a duplex assay kit (Advanced Cell Diagnostics Inc., Newark, CA, USA). The RNA-ISH targets were the rep gene of PCV2 and PCV3 (Genbank: KX298474.1 and HQ839721.1, respectively). Briefly, slides were incubated for 1 h at 60 °C for paraffin melting, and then dehydrated and re-hydrated in a series of xylene and ethanol washes. The slides were treated with H_2_O_2_ for 10 min, boiled with target retrieval solution for 15 min and incubated with protease at 40 °C for 30 min. Slides were hybridized with the probes at 40 °C for 2 h and after that, a series of amplifications of 30 or 15 min were performed. Hybridization signals for PCV3 and PCV2 were detected as red and green colorimetric staining respectively, using Fast Red and Fast Green solutions. Slides were counterstained with hematoxylin. Non-specific and cross-reaction signals were assessed using known positive and negative slides from a previous case with both PCV3 and PCV2 ISH positive signs and a case without any PCV3 and PCV2 ISH signs, respectively. The microscopic assessment of ISH-RNA results was conducted by two independent blinded pathologists.

Cases were categorized in three groups according to the Ct (cycle threshold) values of PCV3 real-time PCR as the following: strongly positive (Ct ≤ 20), moderately positive (20 < Ct ≤ 30), and mildly positive (30 < Ct < 40). Histopathology results were compiled according to the major lesions observed in each tissue. RNA-ISH slides were classified as positive (+) or negative (−) for PCV2 and PCV3. PCV3-ISH-RNA-positive signals were further classified according to their distribution (focal, multifocal or diffuse) and intensity (mild, moderate or marked) [13].

## 3. Results

A total of 25 cases were selected from the UMN-VDL database. Ten cases were classified as highly positive for PCV3 (Ct ≤ 20, 40%), 12 cases were classified as moderately positive for PCV3 (20 < Ct ≤ 30, 48%), and three cases were classified as mildly positive for PCV3 (30 < Ct < 40, 12%) based on the real-time PCR results for PCV3 on the tissue homogenate samples (Table 2). Detailed information regarding the number and type of tissues in each case is presented in Appendix A.

Heart lesions were observed in 16 out of 21 cases in which heart tissues were submitted. Nine of these 16 cases had cardiac lesions characterized as lymphoplasmacytic myocarditis (Table 1, Figure 1a). Twenty-one out of 25 cases had lung tissues submitted. From those 21, 10 had lymphoplasmacytic interstitial pneumonia (Table 1, Figure 2a). Lymphoplasmacytic periarteritis was the most frequently observed lesion in the kidney samples and in lymph node samples (Table 1, Figure 3a). Lymphoplasmacytic periarteritis was also observed in liver and pancreas (Figure 5a) and spleen samples (Figure 4a). Other less frequent histopathological lesions are presented in Table 1.

ISH-RNA positive signals were classified according to the distribution (focal, multifocal and diffuse) and the intensity (mild, moderate and marked) (Table 3). The tissue with the most frequent PCV3-ISH-RNA positive signals was the heart, where PCV3-ISH-RNA positive signals were observed in cardiomyocytes (20/21) (Figure 1b). Lung with lymphoplasmacytic interstitial pneumonia was the second organ more frequently detected with PCV3-ISH-RNA positive signals, observed mainly in the alveolar septa (18/21) (Figure 2b). The other organs with PCV3-ISH-RNA positive signals were kidney (9/13), liver (9/11), spleen (10/12) (Figure 4b) and lymph node (5/6) (Figure 3b). The distribution of the PCV3-ISH-RNA positive signals (Table 3) was predominantly multifocal (72/84). The intensity of the signals was classified as marked in cardiomyocytes (10/19), arteries of kidneys (5/5), spleen (3/6) and lymph nodes (3/3). The intensity of the PCV3-ISH-RNA positive signals was classified as moderate in spleen white pulp (5/10), lymph node lymphoid follicle (3/5), arteries in the heart (6/13) and pulmonary arteries (4/9). The intensity of the PCV3-ISH signal was classified as mild only in lung alveolar septa (9/18) and kidney (6/9). All the tissues with PCV3-ISH-RNA positive signals also presented positive signals associated to the wall of blood vessels (Figure 5b). PCV3-ISH-RNA positive signals were present in lymphoid follicle even without a histologic lesion in this site (Figure 3a,b).

## 4. Discussion

In the present study, we analyzed cases from which PCV3 infection was confirmed by real-time PCR. Using these cases, we described the association of histological lesions in various tissues with the detection of transcriptionally active virus through the detection of PCV3 mRNA within lesions, supporting the hypothesis that PCV3 is an important swine pathogen. Therefore, the findings of the present study, as obtained from naturally infected pigs, can substantially contribute to the understanding of PCV3 infection.

From the 21 samples of heart, 10 samples belonged to cases classified as highly positive by real-time PCR (Ct of tissue homogenate <20). Within these 10 cases, nine were positive for PCV3 by ISH-RNA. This tropism of PCV3 to cardiac tissues has not been reported for PCV2 in pigs, except in fetus and stillborn piglets [25], but is a common finding in naturally PCV3-infected pigs from the present study and in experimentally PCV3-infected pigs [11,19], which reinforces some distinction on the tissue tropism between PCV3 and PCV2.

Another consistent finding in the present study and in the studies with CD/CD pigs experimentally infected with PCV3 [11,19] was the systemic lymphoplasmacytic periarteritis associated with ISH-RNA positive signals. PCV2 infections have also been reported to cause lymphoplasmacytic vasculitis [26] and ISH-RNA for PCV2 revealed positive signals not only within the inflammatory infiltrate, but also in the cytoplasm of endothelial cells, along with fibrinoid necrosis of arteries. Although in the present study fibrinoid necrosis of arteries was not observed, PCV3-ISH-RNA positive signals were found in association with the inflammatory infiltrate and in blood vessels without inflammation, a fact that indicates the tropism of PCV3 to this site regardless of the presence of inflammatory cells.

Interstitial pneumonia is another common condition between PCV2 and PCV3 infections. In the present study, even in samples classified as moderately- and weakly-positive, PCV3 mRNA was consistently detected in lung sections. The presence of PCV3 mRNA in alveolar septa in cases with interstitial pneumonia in the present study resemble the well described interstitial pneumonia caused by PCV2 [27] and also deserves further investigation.

Finally, lymphoid depletion, one of the most remarkable lesions associated with PCV2 infection, was not a frequent finding associated with PCV3 detection in the present study. Although lymphoid depletion was observed in lymph nodes (1/3) and spleen (2/5), the PCV3 ISH-RNA-positive signals were observed mostly in the periphery of the lymphoid follicles, while lymphoid depletion was associated mostly with PCV2-positive signals. As the PCV2-PCV3 co-infections were more frequently observed in the lymph nodes, we consider lymph node tissue an important candidate for ISH-RNA evaluation when PCV2 and PCV3 are suspected etiologic agents of clinical signs in pigs. PCV2 is known to lead to immunosuppression in infected pigs, and, as a result, PCV2-infected pigs are more susceptible to opportunistic infections [14]. As 20 out of 84 tissue sections analyzed in the present study were ISH-RNA-positive for both PCV2 and PCV3, pig producers need be cautious to possible co-infections with other agents that could aggravate the clinical signs of PCV3 infections.

Since the first report of PCV3 in 2016 [1], an increasing number of cases have been submitted to the UMN/VDL for PCV3 PCR testing based on a variety of clinical presentations, including wasting, dyspnea, weight loss and diarrhea. This scenario reflects the concerns from swine veterinarians and the potential impact of PCV3 in the swine industry. PCV3 was firstly identified by next-generation sequencing in samples from pigs with clinical signs that were compatible with PCV2 infection but negative for PCV2 and other known agents by various ancillary tests [1].

The actual relevance of the PCV3 in the swine industry is still unclear, since the only report of viral isolation and experimental infection in pigs had no animals showing clinical signs in the study [19]. The reproduction of the clinical signs of PCV3 related-diseases in experimental conditions is certainly going to be a hallmark on the understanding of the pathogenesis of PCV3 infection. However, to date, the presentation of clinical signs under experimental or field conditions has been very dissonant. The role of co-infections, immunosuppression and environmental conditions can certainly interfere in the manifestation of clinical signs in PCV3-infected pigs.

## 5. Conclusions

The present study reports a comprehensive investigation of histopathological changes from naturally PCV3-infected pigs and the detection of PCV3 mRNA by in situ hybridization. The most frequent histological lesions were present in heart and lung and were characterized as lymphoplasmacytic myocarditis and lymphoplasmacytic interstitial pneumonia, respectively, with lesions coinciding with ISH-RNA positive signals for PCV3. The tropism of PCV3 to cardiac tissues reinforces a distinction on tissue tropism between PCV3 and PCV2. Although heart and lung tissues showed to be important for PCV3 diagnosis, lymphoid tissues (i.e., spleen and lymph nodes) are essential for differential diagnosis, specially related with PCV2-associated diseases. Taken together, the findings of this study not only serve as a guide for sampling in cases of PCV3-suspected disease, but also provide data to the understanding to this relatively new virus of swine production.

## Figures and Tables

**Figure 1 animals-11-01520-f001:**
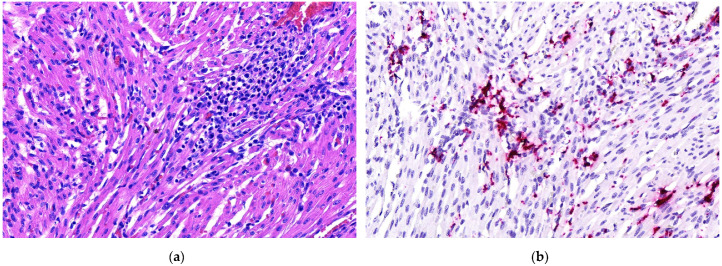
Histopathological lesions associated with mRNA in situ hybridization (ISH-RNA) positive signals for porcine circovirus type 3. (**a**) Heart (20×). Lymphoplasmacytic myocarditis, HE. (**b**) Heart (20×). Diffuse, marked PCV3-ISH-RNA positive signals.

**Figure 2 animals-11-01520-f002:**
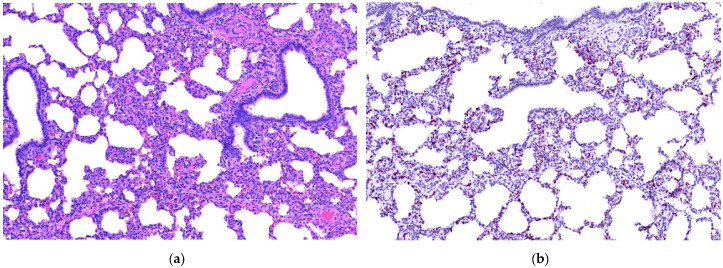
Histopathological lesions associated with mRNA in situ hybridization (ISH-RNA) positive signals for porcine circovirus type 3. (**a**) Lung (10×). Lymphoplasmacytic interstitial pneumonia, HE. (**b**) Lung (10×). Diffuse, marked PCV3-ISH-RNA positive signals.

**Figure 3 animals-11-01520-f003:**
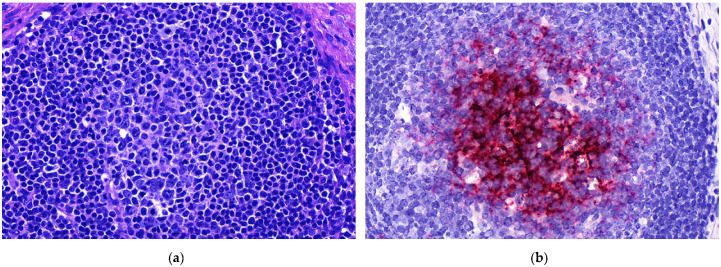
Histopathological lesions associated with mRNA in situ hybridization (ISH-RNA) positive signals for porcine circovirus type 3. (**a**) Lymph node (20×), apparently normal lymphoid follicle, HE. (**b**). Lymph node (20×), lymphoid follicle. Focal, marked PCV3-ISH-RNA positive signals.

**Figure 4 animals-11-01520-f004:**
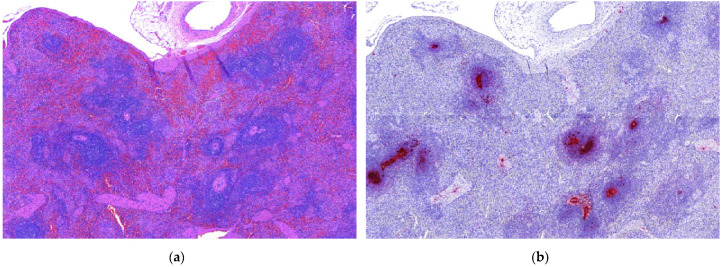
Histopathological lesions associated with mRNA in situ hybridization (ISH-RNA) positive signals for porcine circovirus type 3. (**a**) Spleen (4×). Lymphoid depletion and periarteritis, HE. (**b**) Spleen (4×). Multifocal, marked PCV3-ISH-RNA positive signals in the center of lymphoid follicles.

**Figure 5 animals-11-01520-f005:**
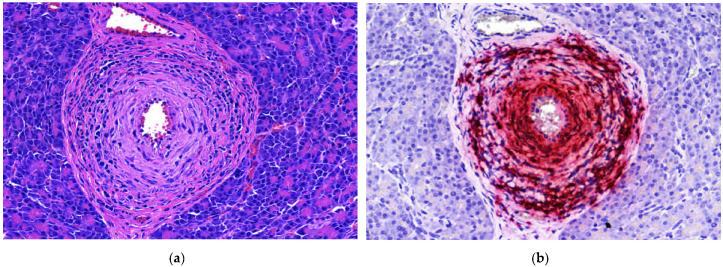
Histopathological lesions associated with mRNA in situ hybridization (ISH-RNA) positive signals for porcine circovirus type 3. (**a**) Pancreas, artery (20×). Lymphoplasmacytic periarteritis, HE. (**b**) Pancreas, artery (20×). Focal, marked PCV3-ISH-RNA positive signals.

**Table 1 animals-11-01520-t001:** Histopathological findings from real-time PCR PCV3-positive cases.

Organ	No. of Samples with Lesions	Most Common Histologic Lesion
Heart	16/21	myocarditis (9/16)
		epicarditis (6/16)
Lung	14/21	interstitial pneumonia (10/14)
		bronchopneumonia (3/14)
Kidney	3/13	periarteritis (2/3)
		glomerulonephritis (1/3)
Liver	2/11	periarteritis (1/2)
		hepatitis (1/2)
Spleen	5/12	lymphoid depletion (2/5)
		lympholysis (2/5)
		periarteritis (2/5)
Lymph node	3/6	periarteritis (2/3)
		lymphoid depletion (1/3)

**Table 2 animals-11-01520-t002:** Real-time PCR findings for PCV3 in naturally infected pigs.

Classification According to Real-Time PCR Ct	Number of Samples
PCV3 highly positive (Ct ≤ 20)	10/25
PCV3 moderately positive (20 < Ct ≤ 30)	12/25
PCV3 mildly positive cases (30 < Ct < 40)	3/25

**Table 3 animals-11-01520-t003:** Histological distribution and intensity of PCV3-ISH-RNA positive signals and PCV2 co-infection detected by ISH-RNA.

Tissue	PCV2 by ISH-RNA	Site of PCV3-ISH-RNA Signal	No. of PCV3-ISH-RNA Positive Samples	Distribution	Intensity
Focal	Multifocal	Diffuse	Mild	Moderate	Marked
Heart	7/21	Cardiomyocytes	20/21	2	11	8	6	5	10
Arteries	13/21	1	11	1	4	6	3
Lung	3/21	Alveolar septa	18/21	1	9	8	9	3	6
Arteries	9/21	0	8	1	3	4	2
Bronchioles	3/21	2	1	0	3	0	0
Kidney	2/13	Parenchyma	10/13	1	7	1	6	2	1
Arteries	5/13	0	3	2	0	0	5
Liver	2/11	Parenchyma	9/11	1	5	3	6	0	3
Arteries	4/11	1	2	1	1	1	1
Spleen	4/12	White pulp	10/12	0	10	0	1	5	4
Red pulp	5/12	0	4	1	2	2	1
Arteries	6/12	0	6	0	1	2	3
Lymph node	2/6	Lymphoid follicle	5/6	0	4	1	1	3	1
Parenchyma	1/6	0	1	0	0	0	1
Arteries	3/6	1	1	1	0	0	3
Total	20/84		72/84

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
