# Peer review of "Histological Lesions and Replication Sites of PCV3 in Naturally Infected Pigs"

_animals, 2021, doi:10.3390/ani11061520_

Round 1

Reviewer 1 Report

This is a nice study of tissue already collected and determined to be PCR positive for PCV3.  The samples have been well prepared for H&E and imaged well, the conclusions are supported by the results.  I only have a few typos which need to be addressed, they are attached in a Word document

Author Response

Reviewer’s comments (authors’ replies highlighted in blue)

Page 1, line 12 ‘….is a challenge…’

This has been changed.

Page 2, line 48 ‘…and has less than…’

This has been changed.

Page 2, line 65 ‘…the clinical signs of PCV3 associated disease were not observed…’

This sentence was removed based on comments to reduce the introduction section for another reviewer.

Page 2, line 77 ‘…development of comprehensive diagnostic criteria…’ (remove ‘a’)

This has been changed.

Page 3, line 123 ‘naturally’

This has been changed.

Page 3, line 126 ‘naturally’

This has been changed.

Page 4, line 143 ‘artery’

This has been changed.

Page 7, line 221 ‘…However, to date, the presentation…’ (remove ‘up’)

This has been changed.

Page 7, line 222 ‘…conditions has been very…’

This has been changed.

Page 7, line 223 ‘…immunosuppression and environmental conditions…’

This has been changed.

Reviewer 2 Report

The study evaluated PCV3 in situ hybridization (ISH) assay together with histological lesions and real time qPCR in naturally infected pigs. The authors performed the ISH test on different tissues from 25 PCV3 total cases selected for their positivity by real time qPCR and for the presence of 3 formalin-fixed sections of the following organs: heart, spleen, lung, kidney, liver and/or lymph nodes. The main histological lesions associated with the presence of PCV3 mRNA were lymphoplasmacytic inflammatory infiltrates in tissues, in particular lymphoplasmacytic myocarditis and epicarditis, and lymphoplasmacytic interstitial pneumonia. Authors suggested that the sampling of heart and lung is critical for PCV3 diagnosis by real time qPCR and ISH.

The paper has interesting findings but it is affected by lack of clear information and poor layout. Mismatch in sample information and the lack of relevant figures and references are limiting the manuscript. Additionally, it needs some level of English revision. For example, some mistyping are present in the text. E.g. line 12 “a” challenge line 123, 126 naturally etc.

The introduction should be more focused on PCV3 literature and less on PCV2.

Despite the description of the case selection, it is still not clear to the reviewer why authors have selected 25 cases and how. Which tissue was used to define the ‘highly positivity” of the qPCR data? 25 cases were selected out of how many qPCR +?

Do authors consider 25 cases a significant number for the final evaluation?

Additionally, authors stated that heart and lung are critical tissues for PCV3 diagnosis (as stated in the conclusion and partially in the abstract) but no H-E and ISH figures of those tissues have been included in the manuscript (only liver and lymph nodes). On the other side, lymph nodes figures (Fig. 1-2) have been included but are not listed as relevant tissues for diagnosis. Moreover, lymph nodes are only 6 tissues out of 84 as listed in Suppl. Table 1.

The reviewer consider lymph nodes a very important tissue for PCV infections and pathogenesis (both PCV2 and PCV3), mainly when both viruses can be detected in the same samples. In fact, most of the coinfections were detected in lymph nodes as stated in lines 148 “The co-infection was mostly observed in lymph nodes, especially in the lymphoid 148 follicles (2/6)”.Author should acknowledge the relevance of this tissue or discuss why they consider it not relevant for diagnosis.

Minor comments:

No match between samples in the manuscript: 87 or 86 or 84? 87 line 147 no match with Table 1 (86) and Suppl table 1 (84) and table 4 (84) line 206. Please clarify these differences or revise the information.

Please remove 16/21, 9/16 and 6/16 from the abstract. The statement of % is enough to summarize the outcome. Add % of lymphoplasmacytic pneumonia in the abstract, removing 21 and 10 values.

The reviewer considers that myocarditis does not fit in the keywords.

Line 22. Authors have not considered that one successful experimental infection model has been published, see ref:

Induction of Porcine Dermatitis and Nephropathy Syndrome in Piglets by Infection with Porcine Circovirus Type 3. Jian et al. J Virol 2019 Feb 5;93(4):e02045-18.doi: 10.1128/JVI.02045-18. Print 2019 Feb 15. PMID: 30487279   PMCID: PMC6363995  DOI: 10.1128/JVI.02045-18

Missing reference for line 38-39 statement related to the identification of PCV3 in US.

Line 40: authors used 3 refs for China PCV3 when only one is really needed to support the statement.

Line 43 missing reference for statement related to PCV2 clinical signs

Line 46: the reviewer do not consider Ref 14 the best citation for the sentence. A review on porcine circovirus associated diseases would be more appropriated.

For example, a recent review have been published on the topic: “Porcine circoviruses: current status, knowledge gaps and challenges” Opriessnig et al. Virus Research 2020. https://doi.org/10.1016/j.virusres.2020.198044

Authors need to check the format for references. In some cases is not adequate (see reference list, #15)

Line 81 It is not clear to the reviewer if the PCR performed is a semi-quantitative PCR or a real time qPCR. The information needs to be consistent along the manuscript. See line 81 and line 110.

Line 62 generally references are included at the end of the sentences - change refs 20,13

Line 64-66 and 74-77 Authors cited the same reference and express a similar concept. Both sections need a better summary. Moreover, the reviewer think that there is a lack of good experimental models for PCV3 but the mechanism of pathogenesis is quite similar to other PCV, therefore, not completely unclear. Additionally, two experimental infections were performed, one successful (see Jian et al.  JVI 2019) and one without clinical signs (Mora-Diaz et al. Viruses 2020).

Line 86 Failure to thrive. The reviewer would suggest the word “wasting”

Line 87 specify real time qPCR

Line 88 Ct of 40 is generally a “negative case”, not a ”positive case”. Please add the Ct related to the detection limit

Line 95 need reference to Table 3

Line 99 60°C

M&M. Describe the positive and negative controls for the PCV3 ISH.

Authors need to show or add a reference for the different intensity of the positive signals. It would be good to have Suppl. Figures with representative figures of the 3 intensities. The same applies for distribution.

The reviewer consider that it would be more helpful if references to tables and figures are in bold along all the manuscript.

Table 1. PCV3 and PCV2 coinfections were detected by qPCR and they should be included in Table 2 instead. In fact, Table 1 title describe the selection for ISH samples.

The section in lines 127-135 is just a repetition of the Table 3. Please summarized the section only with relevant information not contained in the table.

Figure 1-3 Add magnification in the legend and bar in the figures.

Figure 1  a)…. b)….. Figure 2 a)… b)…

Line 148 Add figure of ISH coinfection in lymph node as it is a relevant finding. If not the statement should be considered as “data not shown”. “PCV2/PCV3 coinfection was observed in 20 of a total of 87 tissue samples analyzed 147 (23%) (data not shown)” .

Add the information that PCV2/PCV3 coinfection was observed in 20 of a total of 87 tissue samples analyzed 147 (23%) in Table 1

Line 158 “ positive signals (table 3)”  should be (Table 4) instead

Table 4. Title of the table is not clear.

No. of positive samples of PCV3?

PCV2 infection detected by ISH; number are not lined with the tissue so it is not clear if the number is general or is related to a specific location. It would be helpful if also PCV2 positivity is lined with the tissue where detected.

Line 187 in situ

Line 190 ISH not ISHA

Line 192 tropism of PCV3

Author Response

Reviewer’s comments (authors’ replies highlighted in bold)

The study evaluated PCV3 in situ hybridization (ISH) assay together with histological lesions and real time qPCR in naturally infected pigs. The authors performed the ISH test on different tissues from 25 PCV3 total cases selected for their positivity by real time qPCR and for the presence of 3 formalin-fixed sections of the following organs: heart, spleen, lung, kidney, liver and/or lymph nodes. The main histological lesions associated with the presence of PCV3 mRNA were lymphoplasmacytic inflammatory infiltrates in tissues, in particular lymphoplasmacytic myocarditis and epicarditis, and lymphoplasmacytic interstitial pneumonia. Authors suggested that the sampling of heart and lung is critical for PCV3 diagnosis by real time qPCR and ISH. The paper has interesting findings, but it is affected by lack of clear information and poor layout. Mismatch in sample information and the lack of relevant figures and references are limiting the manuscript. Additionally, it needs some level of English revision. For example, some mistyping are present in the text. E.g. line 12 “a” challenge line 123, 126 naturally etc.

The introduction should be more focused on PCV3 literature and less on PCV2.

The authors reviewed and reduced the introduction as requested.

Despite the description of the case selection, it is still not clear to the reviewer why authors have

selected 25 cases and how. Which tissue was used to define the ‘highly positivity” of the qPCR

data? 25 cases were selected out of how many qPCR +?

Cases were selected based on the tissue submissions received in the UMN-VDL where the diagnostic report indicated a potential involvement of PCV3 in the clinical outcome. There are mainly 2 reasons we didn’t focus on reporting the total numbers of PCV3 qPCR: (i) the UMN lab runs PCV2 and PCV3 qPCR in virtually all the cases submitted with suspect of systemic disease and (ii)  the diagnostic report based on clinical history and histological lesions were the starting point to select the case. We’ve added a sentence to clarify this in the “case selection” in the material and methods.

Do authors consider 25 cases a significant number for the final evaluation?

For descriptive purposes and the intention of contributing for scientific community on the most common PCV3-associated lesions, we believe 25 cases were a reasonable number to show consistence.

Additionally, authors stated that heart and lung are critical tissues for PCV3 diagnosis (as stated

in the conclusion and partially in the abstract) but no H-E and ISH figures of those tissues have

been included in the manuscript (only liver and lymph nodes). On the other side, lymph nodes

figures (Fig. 1-2) have been included but are not listed as relevant tissues for diagnosis.

This was an error. Lymphoid tissues were added as important sample.

Moreover, lymph nodes are only 6 tissues out of 84 as listed in Suppl. Table 1.

The reviewer consider lymph nodes a very important tissue for PCV infections and

pathogenesis (both PCV2 and PCV3), mainly when both viruses can be detected in the same

samples. In fact, most of the coinfections were detected in lymph nodes as stated in lines 148

“The co-infection was mostly observed in lymph nodes, especially in the lymphoid 148 follicles

(2/6)”.Author should acknowledge the relevance of this tissue or discuss why they consider it

not relevant for diagnosis.

We’ve acknowledged this point and we’ve added it the conclusions.

Minor comments:

No match between samples in the manuscript: 87 or 86 or 84? 87 line 147 no match with Table

1 (86) and Suppl table 1 (84) and table 4 (84) line 206. Please clarify these differences or revise

the information.

Please remove 16/21, 9/16 and 6/16 from the abstract. The statement of % is enough to

summarize the outcome. Add % of lymphoplasmacytic pneumonia in the abstract, removing 21

and 10 values.

This has been removed.

The reviewer considers that myocarditis does not fit in the keywords.

The word was removed from the keywords.

Line 22. Authors have not considered that one successful experimental infection model has

been published, see ref:

Induction of Porcine Dermatitis and Nephropathy Syndrome in Piglets by Infection with Porcine

Circovirus Type 3. Jian et al. J Virol 2019 Feb 5;93(4):e02045-18.doi: 10.1128/JVI.02045-18.

Print 2019 Feb 15. PMID: 30487279 PMCID: PMC6363995 DOI: 10.1128/JVI.02045-18

Missing reference for line 38-39 statement related to the identification of PCV3 in US.

This reference was added in the introduction.

Line 40: authors used 3 refs for China PCV3 when only one is really needed to support the

statement.

This has been changed.

Line 43 missing reference for statement related to PCV2 clinical signs

This has been added.

Line 46: the reviewer do not consider Ref 14 the best citation for the sentence. A review on

porcine circovirus associated diseases would be more appropriated.

For example, a recent review have been published on the topic: “Porcine circoviruses: current

status, knowledge gaps and challenges” Opriessnig et al. Virus Research 2020.

https://doi.org/10.1016/j.virusres.2020.198044

This reference has been added.

Authors need to check the format for references. In some cases is not adequate (see reference

list, #15)

The format for references was revised and adjusted.

Line 81 It is not clear to the reviewer if the PCR performed is a semi-quantitative PCR or a real

time qPCR. The information needs to be consistent along the manuscript. See line 81 and line

110.

This has been changed.

Line 62 generally references are included at the end of the sentences - change refs 20,13

This has been changed.

Line 64-66 and 74-77 Authors cited the same reference and express a similar concept. Both

sections need a better summary. Moreover, the reviewer think that there is a lack of good

experimental models for PCV3 but the mechanism of pathogenesis is quite similar to other

PCV, therefore, not completely unclear. Additionally, two experimental infections were

performed, one successful (see Jian et al. JVI 2019) and one without clinical signs (Mora-Diaz

et al. Viruses 2020).

This section has been modified and edited as suggested.

Line 86 Failure to thrive. The reviewer would suggest the word “wasting”

This word has been replaced.

Line 87 specify real time qPCR

The real time qPCR criteria further clarified in the first paragraph of the results.

Line 88 Ct of 40 is generally a “negative case”, not a ”positive case”. Please add the Ct related

to the detection limit

This was removed from the M&M. A better description was made in the first paragraph of the result’s section.

Line 95 need reference to Table 3

This was added.

Line 99 60°C

M&M. Describe the positive and negative controls for the PCV3 ISH.

Authors need to show or add a reference for the different intensity of the positive signals. It

would be good to have Suppl. Figures with representative figures of the 3 intensities. The same

applies for distribution.

A reference based on PCV2 IHC grading system was added, since there isn’t reference for PCV3 to date.

The reviewer consider that it would be more helpful if references to tables and figures are in

bold along all the manuscript.

This was been added.

Table 1. PCV3 and PCV2 coinfections were detected by qPCR and they should be included in Table 2 instead. In fact, Table 1 title describe the selection for ISH samples.

For clarity, this table was removed. The information on this table has been already described either in Table 2 or in the M&M.

The section in lines 127-135 is just a repetition of the Table 3. Please summarized the section

only with relevant information not contained in the table.

The repetitive results from the table was removed and summarized.

Figure 1-3 Add magnification in the legend and bar in the figures.

Figure 1 a)…. b)….. Figure 2 a)… b)…

The magnification was added in the figure’s description. As recent work from the journal haven’t add a bar in the figures (De las Heras, 2021), the authors understood that the magnification was enough. To add the bars, the authors would have to take new pictures. Please inform us if necessary. 

Line 148 Add figure of ISH coinfection in lymph node as it is a relevant finding. If not the

statement should be considered as “data not shown”. “PCV2/PCV3 coinfection was observed in

20 of a total of 87 tissue samples analyzed 147 (23%) (data not shown)” .

This has been added.

Add the information that PCV2/PCV3 coinfection was observed in 20 of a total of 87 tissue

samples analyzed 147 (23%) in Table 1

For better clarity and to avoid repetition this information was described in the text and the Table 1 was removed.

Line 158 “ positive signals (table 3)” should be (Table 4) instead

This has been changed.

Table 4. Title of the table is not clear.

No. of positive samples of PCV3?

PCV2 infection detected by ISH; number are not lined with the tissue so it is not clear if the

number is general or is related to a specific location. It would be helpful if also PCV2 positivity is

lined with the tissue where detected.

The title of the table was edited, and the table was modified as suggested.

Line 187 in situ

This has been changed.

Line 190 ISH not ISHA

This has been changed.

Line 192 tropism of PCV3

This has been changed.

Round 2

Reviewer 2 Report

Authors have considerably improved the manuscript and can be accepted after minor changes.

Minor comments:

Line 30 “ lymph nodes”

Line 57 “diseases is still..”

Line 66 remove “were” from were observed

Line 67 “lesions”

Line 68 The reviewer suggests “a” instead of "with"

Line 87 “based on tissue"

Line 88 “suggested”

Table 2. “naturally”   Number of “cases”   

                  remove “cases” in PCV3 highly positive and moderately positive or add "cases" in mildly positive

line 139 in situ

 Figure 3a-3b  Liver, artery (20x)

The reviewer consider the previous figures Figure 1a (Lymph node. Lymphoid depletion and periarteritis, HE) and Figure 1b (Lymph node. Multifocal, marked PCV3-ISH-RNA positive signals in the center of the lymphoid follicle), from the original version, are very relevant for the paper.

The reviewer suggests to add those figures again in the main figures section (if possible) or as supplementary material, with magnification.

Line 173 ISH-RNA to be consistent with the rest of the manuscript.

Line 185-189 This is a very significant outcome, so the reviewer considers it would be relevant to highlight it even more in the conclusion.

Line 239 “important for PC3”

Regarding the use of a bar in the figures, the reviewer accepts what are the standards of the journal.
